# An evaluation of the impact of social and structural determinants of health on forgone care during the COVID-19 pandemic in Baltimore, Maryland

**Diane Meyer**[1,2]*, **Kelly Lowensen**[1], **Nancy Perrin**[3], **Ayana Moore**[4], **Shruti H. Mehta**[5], **Cheryl R. Himmelfarb**[3], **Thomas V. Inglesby**[2], **Jacky M. Jennings**[6], **Alexandra K. Mueller**[6], **Jessica N. LaRicci**[1], **Woudase Gallo**[1], **Adam P. Bocek**[1], **Jason E. Farley**[1]

1 Center for Infectious Disease and Nursing Innovation, Johns Hopkins University, School of Nursing, Baltimore, MD, United States of America, 2 Center for Health Security, Johns Hopkins University, Bloomberg School of Public Health, Baltimore, MD, United States of America, 3 Johns Hopkins University, School of Nursing, Baltimore, MD, United States of America, 4 FHI 360, Durham, NC, United States of America, 5 Department of Epidemiology, Johns Hopkins University, Bloomberg School of Public Health, Baltimore, MD, United States of America, 6 Department of Pediatrics, Johns Hopkins University, School of Medicine, Baltimore, MD, United States of America

* dmeyer10@jhmi.edu

**Data Availability Statement:** All relevant data are within the paper and its Supporting Information files.

## Abstract

Evidence suggests that reductions in healthcare utilization, including forgone care, during the COVID-19 pandemic may be contributing towards excess morbidity and mortality. The objective of this study was to describe individual and community-level correlates of forgone care during the COVID-19 pandemic. We conducted a cross-sectional, secondary data analysis of participants (n = 2,003) who reported needing healthcare in two population-representative surveys conducted in Baltimore, MD in 2021 and 2021–2022. Abstracted data included the experience of forgone care, socio-demographic data, comorbidities, financial strain, and community of residence. Participant's community of residence were linked with data acquired from the Baltimore Neighborhood Indicators Alliance relevant to healthcare access and utilization, including walkability and internet access, among others. The data were analyzed using weighted random effects logistic regression. Individual-level factors found to be associated with increased odds for forgone care included individuals age 35–49 (compared to 18–34), female sex, experiencing housing insecurity during the pandemic, and the presence of functional limitations and mental illness. Black/African American individuals were found to have reduced odds of forgone care, compared to any other race. No community-level factors were significant in the multilevel analyses. Moving forward, it will be critical that health systems identify ways to address any barriers to care that populations might be experiencing, such as the use of mobile health services or telemedicine platforms. Additionally, public health emergency preparedness planning efforts must account for the unique needs of communities during future crises, to ensure that their health needs can continue to be met. Finally, additional research is needed to better understand how healthcare access and utilization practices have changed during versus before the pandemic.

**Funding:** Research reported in this publication was supported by the National Institute of Nursing Research (award number 1F31NR080834-01) and the National Institute of Allergy and Infectious Diseases of the National Institutes of Health under award numbers P30AI094189-09S1 and UM1AI068619. The content is solely the responsibility of the authors and does not necessarily represent the official views of the National Institutes of Health. The funders had no role in study design, data collection and analysis, decision to publish, or preparation of the manuscript.

**Competing interests:** S. Mehta receives material support from Abbott Diagnostics. This does not alter our adherence to PLOS ONE policies on sharing data and materials. The authors do not have any other conflicts of interest to disclose

## Introduction

Evidence suggests that reductions in healthcare utilization during the Coronavirus disease 2019 (COVID-19) pandemic may be contributing towards excess morbidity and mortality [1–3]. Much of this evidence consists of studies describing changes in the volume of services rendered, such as trends in hospital admissions and emergency department usage [1, 4, 5]. Fewer studies have evaluated COVID-19 related changes in healthcare utilization through the lens of patient-reported forgone care, which is defined as healthcare that is perceived as needed by the person but not received, and includes delayed, missed, or skipped visits with a healthcare provider [6, 7].

Existing studies have identified several individual-level social and structural determinants of health (SSDoH) as important correlates of forgone care during the pandemic, although the findings have been at times contradictory, likely related to the heterogeneity of sampling [8–19]. Czeisler et al., using a nationally representative sample of US adults, found the prevalence ratio of forgone care in non-Hispanic Black adults to be 1.6 times that of non-Hispanic White adults in June 2020 [10]. However, another nationally-representative survey of forgone care from March to July 2020 among adults did not identify any statistically significant difference in the frequency of forgone care by race/ethnicity [9]. A separate study that evaluated forgone care among Medicare beneficiaries found a higher prevalence among non-Hispanic White beneficiaries (22.5%), compared to non-Hispanic Black beneficiaries (12.6%) in the summer of 2020, although these differences dissipated when looking at forgone care in the fall of 2020 and winter of 2021 [16]. Other factors identified as being significantly correlated with forgone care, although at times with mixed results, include experiencing food or income insecurity; education level; having comorbid conditions, including mental health conditions and functional disabilities; both older and younger age; having health insurance; identifying as a member of the LGBTQ+ community; and experiencing racial discrimination [8–13, 15–17, 19]. These studies may indicate that certain individuals and communities experienced disproportionate impacts on healthcare access and utilization during the pandemic, leading to forgone care.

Community-specific SSDoH, such as Area Deprivation Index, internet coverage, walkability, and ethnic segregation are related to health outcomes [20–26]. Studies have identified geographic inequities related to the COVID-19 pandemic, including COVID-19 case counts and mortality rates [27, 28] and inequitable access to testing and treatment [29–31], but few have acknowledged these factors and their associations with forgone care. Similar geographic inequities may have occurred in who experienced forgone care, such as in communities with high rates of poverty, those with ethnic segregation, and those without resources such as internet coverage. For example, several community-level factors have been cited in the literature as reasons for forgoing care during the pandemic, including closures of local health facilities, transportation challenges, and lack of resources required for telemedicine [1, 9, 32, 33]. Also concerning is that many of the communities that suffered higher rates of SARS-CoV-2 infection also likely needed additional healthcare services as a result.

Forgone care can have drastic consequences on individual and community health. Delayed and cancelled preventive care, including immunizations, dental cleanings, and cancer screenings, will have long-term health implications that go well beyond the end of the COVID-19 pandemic. For example, one study identified decreases in screenings for breast, lung, colon, cervical, and prostate cancer ranging from 60% to 82% at one large health center in Massachusetts [34]. Delayed treatment for emergent conditions such as strokes, heart attacks, and orthopedic trauma likely have led to worse prognoses and more severe outcomes [35–37]. Furthermore, studies conducted prior to the pandemic found that those with chronic conditions are more likely to forgo care [12]. Given that the pandemic introduced or exacerbated

existing barriers to healthcare access, these individuals could be at higher risk for severe down-stream consequences of interruptions in care. This is reflected in mortality data published by the US Centers for Disease Control and Prevention, which suggests a large increase in excess deaths from chronic conditions such as heart disease and diabetes, highlighting the importance of continued access to care and the potential consequences of forgone care [38].

The aim of this study was to evaluate the impact of individual and community-level SSDoH on healthcare utilization during the COVID-19 pandemic by specifically looking at forgone care among a sample of adults living in Baltimore, Maryland. The objective was to determine whether community-level correlates accounted for forgone care above and beyond any individual participant-level correlates. Understanding the intersections of forgone care during the COVID-19 pandemic with SSDoH is critical to providing a comprehensive view of the health impacts of the pandemic and to identifying which individuals and communities experienced the greatest interruptions in care.

## Methods

### Study design and data sources

This was a secondary data analysis using baseline data abstracted from two parent studies funded by the National Institute of Allergy and Infectious Diseases. The combined analytic sample leveraged the strengths of two robust sampling strategies used in the parent studies, increasing the overall sample size, and helping to ensure adequate representation of economically disadvantaged and historically excluded communities that may not be included in the current COVID-19 forgone care literature.

This nested study used a cross-sectional design to examine individual and community-level factors associated with forgone care during the COVID-19 pandemic in Baltimore, MD. Data abstracted from the parent studies included responses to survey questions regarding demographics, comorbidities, healthcare needs, and forgone care during the COVID-19 pandemic. For the purposes of this study, forgone care was defined as healthcare that was skipped, missed, or delayed during the COVID-19 pandemic. The two parent studies are described in more detail below.

**The Community Collaborative to Combat COVID-19 study.** The Community Collaborative to Combat COVID-19 (C-Forward) parent study was a two-phased comparative effectiveness trial. Enrollment of participants occurred from February 2021 to December 2022. The study aimed to evaluate three different SARS-CoV-2 testing modalities (fixed site, mobile, and home-based) using a representative sample of Baltimore households. The study enrolled households (HH) using a multi-staged sampling approach, organized by census block groups that were stratified by socioeconomic status and race/ethnicity, with oversampling of historically excluded and under-resourced populations (e.g., Hispanic/LatinX populations, low-income whites).

Household recruitment was multi-modal and included doorhangers, mailings, telephone calls, and online methods. The study included enrollment of one eligible member per HH, 18 years of age or older, who was able to provide informed consent (designated as the "HH index member"). The HH index member completed a baseline survey that covered demographics; history of COVID-19 symptoms, testing and treatment; adoption of preventive behaviors; comorbidities and health care access and utilization (including forgone care); COVID-19 impact and pandemic stress; mental health and substance abuse; and knowledge and attitude towards COVID-19. This survey was completed electronically, by mail, or via telephone with a trained research assistant and entered into REDCap. HH index members also received SARS-CoV-2 antigen and antibody testing and other serum testing (e.g., chemistry profiles). A total

of n = 1,978 HH index members were enrolled. Additional information about this study, including other study phases that were not included in this secondary data analysis, can be found on clinicaltrials.gov (clinical trial identifier # NCT04673292).

**The COVID-19 Prevention Network 5002 study.** The COVID-19 Prevention Network 5002 (CoVPN 5002) parent study was a multi-site, cross-sectional SARS-CoV-2 prevalence study conducted from March to July 2021. The study used a venue-time sampling strategy, with random selection of venues and enrollment times. Venues included low-income housing, transitional housing, shelters, soup kitchens, places of worship, local federally qualified health centers, and rehabilitation centers, among others [39].

All individuals accessing the selected venue during the randomly selected time/location slot were approached to assess interest and eligibility to enroll in the study. Adults and children greater than 2 months of age who were willing and able to provide consent (assent for those <18 years of age, with the consent of a guardian) were eligible. Participants completed a paper-based survey with a research assistant that included questions on demographics; medical history (including forgone care); COVID-19 history included exposure; knowledge, attitudes, and behavior about COVID-19; economic impacts of COVID-19; and willingness to take the COVID-19 vaccine. All survey responses were later entered into Medidata. Individuals also received SARS-CoV-2 antigen and antibody testing. A total of n = 1,022 individuals enrolled in the study at the Baltimore, Maryland site. Additional information about this study can be found on clinicaltrials.gov (clinical trial identifier # NCT04658121). The secondary data analysis presented in this manuscript describes only data collected from the Baltimore research site.

## Secondary data analysis

**Inclusion and exclusion criteria.** All parent study participants who reported needing any care type during the COVID-19 pandemic in the parent study survey, including chronic, preventive, or emergent care or a major medical or dental procedure were included (Fig 1); all other parent study participants were dropped (n = 888). Parent study questions can be located in Supplement 1. Participants under the age of 18 were also dropped regardless of whether or not they needed any type of care (n = 22), as C-Forward only included enrollment of HH index members who were ≥ 18 years of age.

Individuals were also excluded if it could not be ascertained as to whether they needed any type of care during the pandemic (e.g., they left these survey questions blank or responses were discordant; n = 44) or it was unclear whether they had experienced the outcome of forgone care (n = 42). Surveys that had seven or more incomplete demographic questions were also dropped (n = 1).

## Measures

**Outcome variable.** The outcome variable for this study was participant-reported forgone care on enrollment in either of the parent studies. This included any forgone care experienced since the beginning of the COVID-19 pandemic, measured from March 1, 2020 until completion of the survey. Questions on forgone care assessed the need for and receipt of chronic or preventive care, emergent care, and major medical or dental procedures. All C-Forward participants responded to questions on needing/forgoing chronic, preventive, and emergent care and elective/dental procedures. For the CoVPN 5002 study, a supplemental survey on needing/forgoing emergent care and elective/dental procedures was added in April 2021 to the Baltimore, MD research site only. Prior to that date, participants only responded to survey questions on needing/forgoing chronic/preventive care.

**Independent variables—Individual level.** The selection of individual-level variables was informed by the Gelberg Behavioral Model for Vulnerable Populations [40]. This framework,

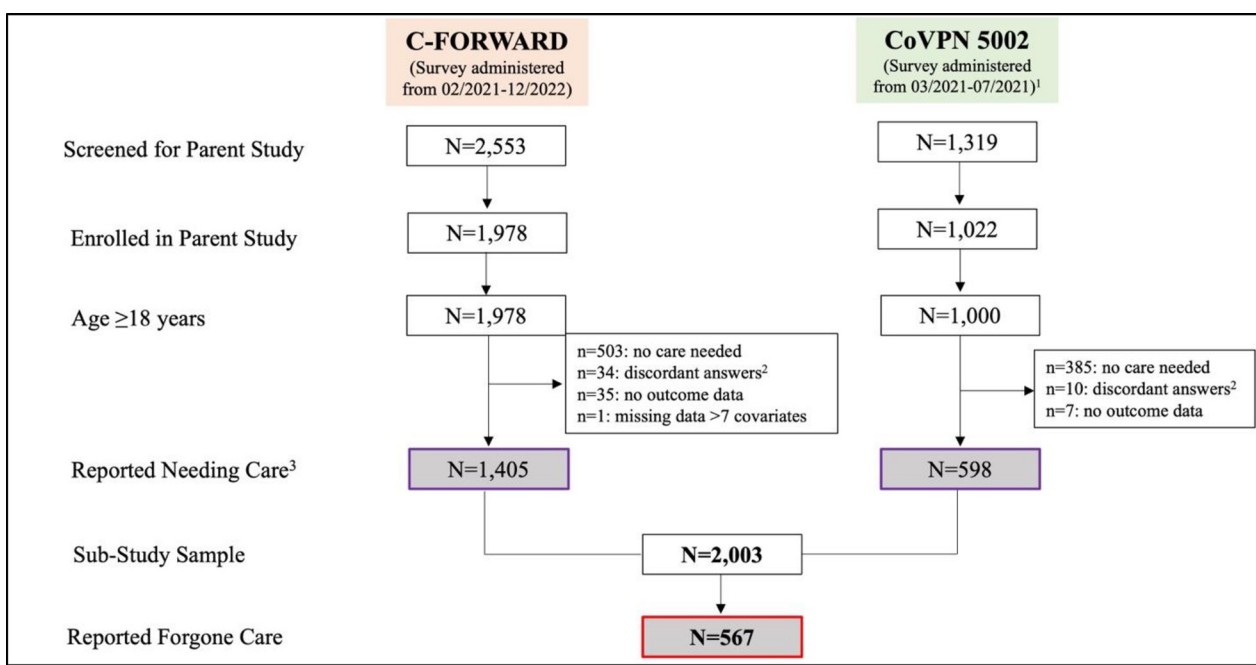

**Fig 1. CONSORT diagram.**

a revision of the Andersen and Newman Model of Healthcare Utilization [41], was chosen as it includes determinants of healthcare system access and utilization relevant to underserved populations living in Baltimore, which has higher rates of poverty, homelessness, crime, and drug use in comparison to other cities of its size as well as widespread disparities across the SSDoH [42–45]. Our adapted model included individual and community-level factors related to healthcare utilization and organizes them into three categories: predisposing—factors that may increase the propensity to seek care; enabling/impeding—factors that may enable or impede use of the health system; and need—factors that reflect perceived need for care [40, 41]. For this study, predisposing factors included age, sex at birth, race, marital status, employment status, disability status, and education level. Enabling/impeding factors included income, loss of housing during the pandemic, and worrying about paying rent/mortgage during the pandemic. Need factors included number of comorbidities, presence of functional limitations, mental illness, and alcohol/substance and injection drug use. Each of these variables were measured using survey data from the two parent studies (see supplement 1 for parent surveys). Responses were consolidated across the two surveys by sociodemographic topic (e.g., sex, age) and sub-categories (e.g., female, male) were created to be inclusive of the wording of the parent studies.

**Independent variables—Community level.** Participants were assigned community-level factors based on their community statistical area (CSA) of residence. CSA's are clusters of neighborhoods that were developed by Baltimore City's Planning Department and are based on city neighborhoods [46]. There are 55 CSA's in the city of Baltimore, with boundaries that align directly to census tracts (46). The population of a given CSA ranges from 5,000 to 20,000 people and consist of anywhere from 1–8 Census Tracts [47]. C-Forward participants were matched to a CSA based on their census block group at enrollment. CoVPN 5002 participants were matched to a CSA based on the address of the venue in which they enrolled, as their actual physical address was not collected for the study.

Seven continuous CSA-level variables that have previously been associated with forgone care (prior to the COVID-19 pandemic) were included in the community-level analysis of forgone care. All CSA-level data was acquired from the Baltimore Neighborhood Indicators Alliance (BNIA) [48]. These variables included the below (year of data collection according to BNIA is notated in parenthesis):

- Percent of households with no internet at home (2020): individuals living in communities with poor internet coverage have been found to be less likely to utilize telehealth services offered during the COVID-19 pandemic [22, 23].

- Walkability score (2017): individuals living in communities with poor walkability have been found to be associated with decreased healthcare access [27]. Challenges associated with poor walkability were likely compounded by the COVID-19 pandemic with closures of public transportation and other nearby resources.

- Percent of population that is Hispanic (2020): segregation of ethnic minorities are associated with "place" disparities, including access to healthcare [24].

- Components of the Area Deprivation Index (ADI): ADI is a census-based index that tracks community socioeconomic status across 17 indicators, including measures of education, employment, quality of housing, and poverty [49]. Higher ADI scores have been previously associated with poorer health outcomes [25, 50]. Unfortunately, ADI scores were unavailable by Baltimore CSA, so 4 related indicators were selected for this analysis that had data available at the CSA-level. These included:

  ○ Percentage of households that pay more than 30% of their total household income on rent and related expenses (2020).

  ○ Percentage of 12th graders in a school year that successfully completed high school (2020/2021).

  ○ Percentage of households whose income fell below the poverty threshold (2020).

  ○ Percentage of population who are not in the labor force (age 16–64 years) (2020).

**Covariates.** Two "pandemic wave" binary variables were added to the models to help account for the different enrollment periods of the parent studies and what waves of the pandemic each participant experienced that may have impacted their ability or desire to seek healthcare. All participants who enrolled on or after August 1, 2021 were coded as having experienced the delta COVID-19 wave; those who enrolled on or after December 1, 2021 were coded as having experienced the omicron COVID-19 wave. A binary variable was also created to account for which parent study the data were acquired from.

## Statistical analyses

**Imputation of missing data.** Patterns of missingness were explored to determine whether data were missing at random. Multiple imputation was then used to impute missing data [51] using Stata version 16.0 (StatCorp, College Station, TX). The sequential imputation using chained equations method was employed for this study, using the individual-level independent variables, the outcome of forgone care, and "study" variable as predictors of missing values. Five total imputed datasets were created. A sensitivity analysis was conducted at the completion of the model-building process to check for biases introduced during multiple imputation. For the sensitivity analysis, the final model was run using only participants with complete data.

The magnitude of the resulting coefficients was then compared with those obtained from running the same model using the imputed dataset.

**Model building—Level-one model.** Weighted bivariate logistic regressions were conducted for each individual-level independent variable, with the binary outcome of any type of forgone care. Each observation was weighted in the regression model for the total amount of years the individual had experienced in the pandemic, measured from March 1, 2020 until the date of survey completion. Any individual-level variable significant at p≤0.10 was carried over to the final level-one multivariable model. The multivariable weighted level-one model also included the "study" and "pandemic wave" variables. Collinearity of independent variables was also checked using variation inflation factor, with none noted.

**Model building—Level-two multi-level model.** The level-two model included weighted random effects logistic regressions with participants clustered within Baltimore CSAs. The model contained all significant level-one variables, and the level-two continuous community-level variables were tested one at a time. Any CSA-level variables significant at p ≤0.10 were incorporated into the final multivariable model. The significance of variables in this final model were set a p≤0.05.

**Power analysis.** A power analysis based on preliminary data assumed an event rate of 0.41 [9, 10], an alpha of 0.05, 80% power, an estimated sample size of n = 1,662, and 30 Baltimore CSAs. Assuming these parameters and an intraclass correlation coefficient (ICC) of 0.010, the study was powered to detect odds ratios that ranged from 1.34 to 1.45, as the distribution of the predictor variable ranged from 20/80 split to a 50/50 split.

## Ethical considerations

This study was approved by the Johns Hopkins University School of Medicine (JHU SOM) Institutional Review Board (IRB) on December 19, 2022 (IRB# 00348695). The C-Forward parent study was approved by the JHU SOM IRB on June 2, 2020 (IRB# 00250298). The CoVPN 5002 parent study was approved by the JHU SOM IRB on January 28, 2021 (IRB#00262004) through a reliance agreement with Advarra IRB, which was the IRB of record for the larger CoVPN 5002 network study. This secondary data analysis was granted a consent waiver as participants who enrolled in either parent study completed consent forms that covered the conduct of this research. During data analysis, authors of this article had access to a limited dataset provided by the parent study data management teams.

## Results

### Participant characteristics

Seventy-one percent of C-Forward participants (1,405 of 1,978) met inclusion criteria for this secondary analysis. A slightly smaller percentage of those enrolled in the CoVPN 5002 study (598 of 1,022; 59%) met inclusion criteria. This yielded a total sample size of n = 2,003 (see Fig 1). Nearly 65% of the sample reported their sex at birth as female. Most (87.6%) reported not being of Hispanic or Latino origin and 52.7% reported their race as Black or African American. Nearly 40% of the sample reported being married or partnered, slightly less than half (45.1%) were employed part or full-time, and 15.5% reported being on disability status. Twenty percent of participants reported being worried about their ability to pay or being unable to pay their rent or mortgage during the pandemic, nearly 8% reported losing their housing during the pandemic, and 30% reported making less than $25,000 per year. Nearly 65% reported having at least one co-morbidity, including a cancer diagnoses, diabetes, renal disease, sickle cell anemia, respiratory disease, cardiac disease, or immune disease (Table 1).

In total, 567 participants reported experiencing any type of forgone care. This included 495 participants who reported forgoing chronic or preventive care, 33 participants who reported forgoing emergent care, and 94 participants who reported delaying an elective surgery or dental procedure. Nearly 57% of those who reported forgone care were over the age of 50 years, 67.9% were female, and over half (50.3%) reported being Black of African American.

## CSA characteristics

The sample included participants from 48 of the 55 CSAs in Baltimore. The number of participants per CSA ranged from 2 to 174. The calculated ICC was 0.09. Table 2 summarizes the mean and standard deviation values for each of the seven CSA-level variables for those who did and did not report forgone care.

## Correlates of forgone care

**Results—Individual-level correlates of forgone care.** Table 3 presents the results from the level-one weighted bivariate analyses examining the association of each individual level variable with forgone care. Several pre-disposing factors were found to be associated with forgone care. Those in the age group 35–49 years were found to have a 1.48 higher odds of forgone care, compared to the age group 18–34 (p = 0.001); conversely, those in the ≥65 age group were found to have a 25% lower odds of forgone care, compared to individuals 18–34 years (p = 0.020). The odds of forgone care were also higher in those reporting female sex at birth (OR 1.24, p = 0.010) and those on disability status (OR 1.43, p = 0.001). Individuals reporting Black or African American race had a 15% lower odds of forgone care (p = 0.047).

Two enabling factors were found to be associated with higher odds of forgone care, including those who reported losing their housing during the COVID-19 pandemic (OR 1.90, p = <0.001) and those who reported being worried about paying their rent or mortgage (OR 1.69, p = <0.001). Finally, significant need level factors with higher odds of forgone care included those with functional limitations (OR 1.66, p<0.001), those with mental illness (OR 1.87, p<0.001), those who reported using substances (OR 1.43, p = 0.004), and those with two or more comorbidities, compared to no comorbidities (OR 1.24, p = 0.023).

**Results—Final multivariable model.** The results of the final weighted multivariable model can be found in Table 3, adjusted for the pandemic wave and study variables and age category, sex, race, disability status, loss of housing, worrying about rent/mortgage, presence of functional limitations, mental illness, substance abuse, and comorbidity number. Predisposing factors that were statistically significant in the final model included those who reported being ages 35–49 years, who were found to have a higher odds of forgone care compared to those in the 18–34 years age group (OR 1.51, p = 0.001). Individuals who reported their sex at birth as female were also found to have higher odds of forgone care, compared to those who reported their sex at birth as male (OR 1.21, p = 0.028). Individuals who reported their race as Black/African American had a 32% reduction in their odds of forgone care, compared to individuals of any other race (OR 0.68, p = ≤0.001). Significant enabling factors included those who reported losing their house during COVID-19, compared to those who did not (OR 1.47, p = 0.014) and those who reported being worried about paying their rent or mortgage, compared to those who were not (OR 1.37, p = 0.005). Need level factors that remained significant in the final model included those with functional limitations, compared to those without (OR 1.46, p = ≤0.001) and those who reported having a mental illness, compared to those without (OR 1.44, p = ≤0.001).

**Results—Community-level correlates of forgone care.** None of the continuous CSA-level variables were found to be significant at the p≤0.10 level after controlling for the

**Table 1. Participant characteristics by forgone care[1].**

| Characteristic | Total Sample (n = 2,003) Number (%) | Reported Any Forgone Care (n = 567) Number (%) | Reported No Forgone Care (n = 1,436) Number (%) |
|---|---|---|---|
| **Age (in years)** | | | |
| 18–34 | 357 (17.8) | 98 (17.3) | 259 (18.1) |
| 35–49 | 420 (21.0) | 148 (26.1) | 272 (18.9) |
| 50–64 | 735 (36.7) | 209 (36.9) | 526 (36.6) |
| ≥65 | 491 (24.5) | 112 (19.8) | 379 (26.4) |
| **Sex at birth** | | | |
| Male | 697 (34.8) | 180 (31.7) | 517 (36.0) |
| Female | 1,291 (64.5) | 385 (67.9) | 906 (63.1) |
| Unknown/PNA | 15 (0.7) | 2 (0.4) | 13 (0.9) |
| **Race** | | | |
| White | 751 (37.5) | 220 (38.8) | 531 (37.0) |
| Black/African American | 1,055 (52.7) | 285 (50.3) | 770 (53.6) |
| Other[2] | 100 (5.0) | 32 (5.6) | 68 (4.7) |
| Unknown/PNA | 97 (4.8) | 30 (5.3) | 67 (4.7) |
| **Ethnicity** | | | |
| Latino/Hispanic origin | 75 (3.7) | 20 (3.5) | 55 (3.8) |
| Not of Latino/Hispanic origin | 1,754 (87.6) | 499 (88.0) | 1,255 (87.4) |
| Unknown/PNA | 174 (8.7) | 48 (8.5) | 126 (8.8) |
| **Married/Partnered** | | | |
| Yes | 783 (39.1) | 219 (38.6) | 564 (39.3) |
| No | 1,184 (59.1) | 343 (60.5) | 841 (58.5) |
| Unknown/PNA | 36 (1.8) | 5 (0.9) | 31 (2.2) |
| **Employed (full or part-time)** | | | |
| Yes | 904 (45.1) | 256 (45.1) | 648 (45.1) |
| No | 1,047 (52.3) | 297 (52.4) | 750 (52.2) |
| Unknown/PNA | 52 (2.6) | 14 (2.5) | 38 (2.7) |
| **Reported being on disability** | | | |
| Yes | 310 (15.5) | 106 (18.7) | 204 (14.2) |
| No | 1,641 (81.9) | 447 (78.8) | 1,194 (83.2) |
| Unknown/PNA | 52 (2.6) | 14 (2.5) | 38 (2.6) |
| **Highest education level** | | | |
| High school or less | 705 (35.2) | 189 (33.3) | 516 (36.0) |
| Some college or more | 1,266 (63.2) | 368 (64.9) | 898 (62.5) |
| Unknown/PNA | 32 (1.6) | 10 (1.8) | 22 (1.5) |
| **Income** | | | |
| <$25,000 | 602 (30.0) | 188 (33.2) | 414 (28.8) |
| $25,000-$49,999 | 306 (15.3) | 71 (12.5) | 235 (16.4) |
| ≥$50,000 | 741 (37.0) | 220 (38.8) | 521 (36.3) |
| Unknown/PNA | 354 (17.7) | 88 (15.5) | 266 (18.5) |
| **Lost housing during the pandemic** | | | |
| Yes | 153 (7.6) | 66 (11.6) | 87 (6.0) |
| No | 1,827 (91.2) | 495 (87.3) | 1,332 (92.8) |
| Unknown/PNA | 23 (1.2) | 6 (1.1) | 17 (1.2) |
| **Worried about or unable to pay rent or mortgage during the pandemic** | | | |
| Yes | 400 (20.0) | 152 (26.8) | 248 (17.3) |
| No | 1,573 (78.5) | 403 (71.1) | 1,170 (81.5) |

*(Continued)*

**Table 1.** (Continued)

| Characteristic | Total Sample (n = 2,003) Number (%) | Reported Any Forgone Care (n = 567) Number (%) | Reported No Forgone Care (n = 1,436) Number (%) |
|---|---|---|---|
| Unknown/PNA | 30 (1.5) | 12 (2.1) | 18 (1.2) |
| **Reported functional limitations[3]** | | | |
| Yes | 616 (30.7) | 218 (38.4) | 398 (27.7) |
| No | 1,361 (68.0) | 337 (59.4) | 1,024 (71.3) |
| Unknown/PNA | 26 (1.3) | 12 (2.1) | 14 (1.0) |
| **Mental illness** | | | |
| Yes | 578 (28.9) | 224 (39.5) | 354 (24.6) |
| No | 1,403 (70.0) | 335 (59.1) | 1,068 (74.4) |
| Unknown/PNA | 22 (1.1) | 8 (1.4) | 14 (1.0) |
| **Substance abuse** | | | |
| Yes | 235 (11.7) | 81 (14.3) | 154 (10.7) |
| No | 1,756 (87.7) | 483 (85.2) | 1,273 (88.7) |
| Unknown/PNA | 12 (0.6) | 3 (0.5) | 9 (0.6) |
| **Number of comorbidities** | | | |
| Zero | 714 (35.6) | 187 (33.0) | 527 (36.7) |
| One | 653 (32.6) | 185 (32.6) | 468 (32.6) |
| Two or more | 636 (31.8) | 195 (34.4) | 441 (30.7) |

[1]Numbers are prior to imputation of missing values

[2]Due to low sample size, individuals reporting their race as American Indian, Native Alaskan, Native Hawaiian, Pacific Islander, or multi-racial were grouped into one category

[3]Functional limitations included physical, mental, or emotional functional limitations

PNA: preferred not to answer

**Table 2. CSA characteristics by forgone care.**

| CSA Characteristic | Total Sample (n = 2,003) Mean [sd] | Reported Any Forgone Care (n = 567) Mean [sd] | Reported No Forgone Care (n = 1,436) Mean [sd] |
|---|---|---|---|
| **Access to internet** | | | |
| % of population with no internet | 16.5 (9.5) | 16.0 (9.3) | 16.7 (9.6) |
| **Walkability** | | | |
| Walkability score | 70.1 (18.6) | 70.5 (18.4) | 69.9 (18.7) |
| **Ethnic makeup** | | | |
| % of population that is Hispanic | 7.7 (9.2) | 8.2 (9.7) | 7.5 (8.9) |
| **Rent affordability** | | | |
| % of HH that pay ≥30% income on rent | 47.6 (8.5) | 47.6 (8.5) | 47.6 (8.5) |
| **High school completion rate** | | | |
| % of 12[th] graders that complete high school | 79.8 (7.7) | 79.9 (7.9) | 79.8 (7.7) |
| **Household poverty rate** | | | |
| % of HH with income ≤ poverty threshold | 16.4 (13.3) | 16.2 (13.3) | 16.5 (13.3) |
| **Unemployment** | | | |
| % of persons not in labor force (age 16–64 years) | 31.1 (11.6) | 31.1 (11.8) | 31.1 (11.5) |

CSA-Community Statistical Area; sd-standard deviation; HH-household

**Table 3. Individual-level correlates of forgone care.**

| Characteristic | Unadjusted Odds of Forgone Care (95% CI) | Adjusted Odds of Forgone Care (95% CI) |
|---|---|---|
| **Predisposing factors** | | |
| **Age (in years)** | | |
| 18–34 | Referent | Referent |
| 35–49 | 1.48 (1.17–1.87)* | 1.51 (1.19–1.93)** |
| 50–64 | 1.04 (0.84–1.29) | 1.05 (0.82–1.34) |
| ≥65 | 0.75 (0.59–0.96)* | 0.80 (0.61–1.04) |
| **Sex at birth** | | |
| Male | Referent | Referent |
| Female | 1.24 (1.05–1.46)* | 1.21 (1.02–1.44)** |
| **Race** | | |
| Any other race | Referent | Referent |
| Black or African American | 0.85 (0.72–1.00)* | 0.68 (0.55–0.83)** |
| **Marital status** | | |
| Not married or cohabitating | Referent | N/A |
| Married or cohabitating | 0.99 (0.85–1.16) | N/A |
| **Employment status** | | |
| Other | Referent | N/A |
| Working full or part time | 1.02 (0.87–1.18) | N/A |
| **Disability status** | | |
| Not on disability status | Referent | Referent |
| On disability status | 1.43 (1.16–1.77)* | 1.13 (0.87–1.47) |
| **Highest education level** | | |
| High school or less | Referent | N/A |
| Some college or more | 1.12 (0.94–1.32) | N/A |
| **Enabling factors** | | |
| **Annual income** | | |
| < $25,000 | Referent | N/A |
| ≥$25,000 | 0.88 (0.75–1.03) | N/A |
| **Lost housing during COVID-19** | | |
| No | Referent | Referent |
| Yes | 1.90 (1.46–2.47)* | 1.47 (1.08–1.99)** |
| **Worried about paying rent/ mortgage** | | |
| No | Referent | Referent |
| Yes | 1.69 (1.41–2.03)* | 1.37 (1.10–1.71)** |
| **Need factors** | | |
| **Functional limitations** | | |
| No | Referent | Referent |
| Yes | 1.66 (1.40–1.96)* | 1.46 (1.20–1.78)** |
| **Mental illness** | | |
| No | Referent | Referent |
| Yes | 1.87 (1.59–2.19)* | 1.44 (1.20–1.73)** |
| **Substance abuse** | | |
| No | Referent | Referent |
| Yes | 1.43 (1.12–1.82)* | 1.08 (0.82–1.43) |
| **Number of comorbidities** | | |
| Zero | Referent | Referent |

*(Continued)*

**Table 3.** (Continued)

| Characteristic | Unadjusted Odds of Forgone Care (95% CI) | Adjusted Odds of Forgone Care (95% CI) |
|---|---|---|
| One | 1.10 (0.92–1.32) | 1.11 (0.92–1.36) |
| Two or more | 1.24 (1.03–1.48)* | 1.24 (1.00–1.55) |

*Denotes those variables in the level one model with p-values ≤0.10 that were carried over to the multivariable model

**Denotes those variables in the multivariable model with p-values ≤0.05

CI: confidence interval

individual level variables of age category, sex, race, disability status, loss of housing, worrying about rent/mortgage, presence of functional limitations, mental illness, substance abuse, and comorbidity number. Thus, none of the CSA-level variables were added to the model with the individual level factors.

## Discussion

Several studies published during the COVID-19 pandemic on forgone care indicate that economically disadvantaged and historically excluded populations experienced a greater prevalence of forgone care [8–10, 12, 14, 19]. However, few studies have investigated correlates of forgone care that go beyond social and economic factors, such as examining the influence of the local community in which people live. Given the widespread impacts of the COVID-19 pandemic on both individuals and communities, it is important to explore the myriad, multilevel SSDoH that may influence forgone care [52]. However, our study found that none of the community-level determinants were significantly associated with forgone care in the multilevel model. This finding differs from previously published literature that has identified community-level disparities in COVID-19-related outcomes, although the geographic unit of analyses used in these studies differ from what was used in our study [27–31].

In this racially and economically diverse sample, we found that Black/African Americans living in Baltimore were less likely to experience forgone care, which is in opposition to some previously published work on forgone care during the COVID-19 pandemic [10, 13, 14]. However, this finding was consistent with Tsuzaki & Taira, who found that non-Hispanic/LatinX Black Medicare beneficiaries were less likely to forgo care in the summer of 2020 compared to White beneficiaries [16]. These findings warrant additional exploration, including comparisons of healthcare utilization practices prior to versus during the pandemic. There are longstanding, systemic inequities in healthcare access and utilization within the Black/African American community that were further exacerbated during the COVID-19 pandemic. For example, a 2016 report by the University of Maryland found that Black/African American Maryland residents were 1.9 times more likely to be unable to afford a visit with a healthcare provider than White residents and were much more likely to die from heart disease, diabetes, stroke, and asthma [53]. Black/African American individuals included in our study sample may have reported less episodes of forgone care because they have historically experienced systemic barriers to access within the healthcare system. Therefore, they may have been less likely to use the healthcare system even prior to the pandemic, and thus less likely to report forgone care.

In our analysis of individual-level correlates of forgone care, we identified those who have experienced housing insecurity during the pandemic, as evidenced by loss of housing or worrying about paying for housing, as having higher odds of forgone care. Income inequities and financial instability have previously been identified as critical determinants of an individual's

healthcare utilization practices and the experience of forgone care [54]. Unfortunately, the pandemic impacted the economic stability of millions of American households, particularly lower-income families, and many lost jobs as a result [55]. This may have drove increased rates of forgone care within individuals experiencing financial hardship and housing instability. Anderson et al., for example, found that 52% of respondents who reported missing medical care did so due to "financial repercussions of the COVID-19 pandemic" [9]. Notably, several safety net programs, such as the Coronavirus Aid, Relief, and Economic Security (CARES) Act, which provided unemployment benefits and additional financial support to American families, and the continuous Medicaid enrollment provision under the Families First Coronavirus Response Act, were available to qualifying individuals early in the pandemic [56, 57]. However, a study published in 2020 on access and enrollment in safety net programs during the pandemic found gaps in awareness for several programs, including health insurance exchanges [58]. Our study also found that individuals reporting female sex at birth had higher odds of forgone care compared to those reporting male sex at birth. During the pandemic, many women had to take on additional caretaking roles, including homeschooling children and caring for ill family members [59]. These additional responsibilities may have impacted their ability to seek healthcare when needed. Further, it was well established prior to the pandemic that men seek healthcare less than women [60], and it is possible this was carried forward in this sample.

The odds of forgone care were also higher in individuals reporting functional limitations. These individuals likely have a higher reliance on support services such as accessible transportation, and interruptions in these services may have disproportionately impacted their ability to access the health system. Those reporting a mental illness also reported an increased odds of forgone care, a finding that aligns with other published COVID-19 forgone care literature [12]. Individuals suffering from mental illness have previously been shown to be at higher risk for forgone care [61], and exacerbations in mental health crises during the pandemic may have disproportionately led to the increased risk of forgone care within this population.

In a world where pandemics and other humanitarian emergencies are increasingly likely due to population growth, interactions at the human-animal interface, climate change, and global travel, it is critical that we strengthen models of care so that health systems can continue to support individual and community health during future public health emergencies. One potential strategy is to increase the use of mobile-based healthcare services. Mobile health clinics have previously been shown to be effective in providing urgent care, preventive health, and chronic disease management and have been shown to increase healthcare access for underserved communities, including the economically disadvantaged [62]. Some of the benefits of mobile health clinics include the elimination of barriers such as lack of transportation, inconvenient operating hours, and long wait times and the ability to form trusting relationships within communities, particularly those who experience stigma such as people living with mental illness [62]. Furthermore, leveraging mobile clinics in the midst of an emergency have been shown to reduce disruptions in care, as they are already well-integrated into the community and can be effectively and quickly deployed [62]. Leveraging of mobile clinics during COVID-19 have been captured in the literature to-date. For example, co-location of COVID-19 vaccines within existing harm reduction mobile services has been demonstrated to be an effective way to deliver vaccines to underserved populations [63]. More widespread use of these types of clinics now and in the future could help reduce forgone care, particularly among those who already experience barriers to healthcare access.

Another strategy that could help address inequities in forgone care during COVID-19 and future health emergencies would be adoption of programs that could extend telehealth services to those who currently lack access. This should include not only the resources required for

telehealth, such as internet coverage and smartphones, but also changes in insurance policies that increase coverage and better education for consumers on how to utilize this technology [64]. For example, participants in one study reported that a major disadvantage of telehealth was their "unfamiliarity" with the platform and distrust that their information was secure and private [65]. Healthcare providers also expressed gaps in knowledge on how to use telemedicine and what services were covered by insurance [65], indicating additional opportunities for education to ensure more consistent and widespread use. Additionally, while important policy changes have been implemented to expand telehealth during COVID-19, permanent adoption of some of these changes, such as compensation for primary healthcare services, could help ensure continued access and would already be in place in the event of a new health emergency [66].

## Limitations

This study has several limitations. First, missing data for participants across several independent variables necessitated imputation of missing values for the model-building process, which may have led to biased results. However, a sensitivity analysis using only participants with complete data revealed similar magnitudes of odds ratios for all variables found to have significance at p≤0.05 in the final, imputed dataset. Second, low sample size amongst participants reporting Hispanic/LatinX origin prohibited incorporation into the final model. Third, the different enrollment periods of the parent studies covered different time periods within the pandemic that may have led to forgone care. We have addressed this limitation by using models weighted by years in the pandemic at time of parent study enrollment. Fourth, because the CoVPN 5002 study did not collect participant's address, we had to assume their residence was within the venue of enrollment. While most venues included low-income housing developments, their assigned CSA may not have been accurately reflected in the data. Fifth, because this study is based upon participant-reported survey data, it may be subject to recall bias. Finally, there is the potential for selection bias which may impact the representativeness of the Baltimore population—for example, the unemployment rate within our sample (52.3%) was higher than that of Baltimore City (29.0%), according to 2021 BNIA data [67], which may be due to greater representation of individuals from CSAs with higher unemployment rates. However, our final study sample was relatively similar across other characteristics, including sex (study sample-54.5% female; Baltimore-53.1% female), race (study sample-52.7% Black/ African American; Baltimore-61.6% Black/African American), ethnicity (study sample-3.7% Hispanic; Baltimore-5.6% Hispanic), and poverty (study sample-30% with income <%25,000 per year; Balitimore-20.3% live below the federal poverty level), when comparing our study sample to 2022 Baltimore census data [42]. Additionally, the CoVPN 5002 parent study did not collect data on participant's health insurance status, which likely impacted healthcare utilization. However, it could be inferred to be similar to the 2022 census data (approximately 6.4% of population under age 65 years is uninsured in Baltimore) [42].

## Conclusions

Despite these limitations, this study has important implications for addressing unmet healthcare needs that have occurred during the pandemic. However, additional research will be critical to further understanding how healthcare access and utilization practices differed during versus before the pandemic and to more clearly identify those who are at highest risk of the downstream consequences of forgone care. It is also critical that the health system identify ways to address any new barriers to care that under-resourced populations might be experiencing, such as the increased use of mobile health services or telemedicine platforms.

Additionally, moving forward, public health emergency preparedness planning efforts must account for the unique needs of these communities during crises, to ensure that their health needs can continue to be met. This will require additional studies that seek to better understand the perspectives and experiences of these populations during the pandemic, and how models of care can be adapted to serve them better in the future.

## Supporting information

**S1 File. Parent study survey questions.**
(DOCX)

## Acknowledgments

We would like to acknowledge the participants of the C-Forward and CoVPN 5002 studies for their participation. We would also like to acknowledge the team at the Center for Infectious Disease and Nursing Innovation for their tireless work in recruitment of participants in both of the parent protocols.

## Author Contributions

**Conceptualization:** Diane Meyer, Nancy Perrin, Ayana Moore, Cheryl R. Himmelfarb, Thomas V. Inglesby, Jason E. Farley.

**Formal analysis:** Diane Meyer.

**Funding acquisition:** Diane Meyer.

**Investigation:** Kelly Lowensen, Jacky M. Jennings, Alexandra K. Mueller, Jessica N. LaRicci, Woudase Gallo, Adam P. Bocek.

**Methodology:** Diane Meyer, Nancy Perrin, Ayana Moore, Shruti H. Mehta, Cheryl R. Himmelfarb, Thomas V. Inglesby, Jason E. Farley.

**Writing – original draft:** Diane Meyer.

**Writing – review & editing:** Diane Meyer, Kelly Lowensen, Nancy Perrin, Ayana Moore, Shruti H. Mehta, Cheryl R. Himmelfarb, Thomas V. Inglesby, Jacky M. Jennings, Alexandra K. Mueller, Jessica N. LaRicci, Woudase Gallo, Adam P. Bocek, Jason E. Farley.

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
