## [Decision Letter · Decision Letter 0]

30 Oct 2023

PONE-D-23-27005Rate of forgone care during the COVID-19 pandemic in Baltimore, MDPLOS ONE

Dear Dr. Meyer,

Thank you for submitting your manuscript to PLOS ONE. After careful consideration, we feel that it has merit but does not fully meet PLOS ONE’s publication criteria as it currently stands. Therefore, we invite you to submit a revised version of the manuscript that addresses the points raised during the review process.

We look forward to receiving your revised manuscript.

Kind regards,

Tae-Young Pak, Ph.D.

Academic Editor

PLOS ONE

Journal Requirements:

 "Research reported in this publication was supported by the National Institute of Nursing Research (award number 1F31NR080834-01, DM; nih.gov) and the National Institute of Allergy and Infectious Diseases of the National Institutes of Health under award numbers P30AI094189-09S1 and UM1AI068619 (JF; nih.gov). The content is solely the responsibility of the authors and does not necessarily represent the official views of the National Institutes of Health."

"I have read the journal's policy and the authors of this manuscript have the following competing interests: S. Mehta receives material support from Abbott Diagnostics. The authors do not have any other conflicts of interest to disclose."

Reviewers' comments:

Reviewer's Responses to Questions

**Comments to the Author**

1. Is the manuscript technically sound, and do the data support the conclusions?

Reviewer #1: Partly

Reviewer #2: No

Reviewer #3: Yes

2. Has the statistical analysis been performed appropriately and rigorously? 

Reviewer #1: No

Reviewer #2: No

Reviewer #3: Yes

3. Have the authors made all data underlying the findings in their manuscript fully available?

Reviewer #1: No

Reviewer #2: Yes

Reviewer #3: Yes

4. Is the manuscript presented in an intelligible fashion and written in standard English?

Reviewer #1: Yes

Reviewer #2: Yes

Reviewer #3: Yes

5. Review Comments to the Author

Reviewer #1: Thank you for allowing me to review this paper. I have major issues with the paper in general, in particular with the methods and analytical pieces of it. I am also not convinced about the novelty and contribution of this study the way the authors frame it. Below are some of my comments:

L41: Studies using data from the Urban Institute capture exactly this question; ie Giannouchos TV, Brooks JM, Andreyeva E, Ukert B. Frequency and factors associated with foregone and delayed medical care due to COVID‐19 among nonelderly US adults from August to December 2020. Journal of evaluation in clinical practice. 2022 Feb;28(1):33-42.

L42: Or choice due to less urgent need – some rate of foregone care might actually be a good thing when patients overutilize the healthcare system. So this statement is partially true and should be revised

L38-46: There are more US studies looking into foregone/delayed care thus the authors should review the literature more careful or at least acknowledge that there is more than “limited” evidence (L43)

A short search revealed the following:

Giannouchos TV, Brooks JM, Andreyeva E, Ukert B. Frequency and factors associated with foregone and delayed medical care due to COVID‐19 among nonelderly US adults from August to December 2020. Journal of evaluation in clinical practice. 2022 Feb;28(1):33-42.

Findling MG, Blendon RJ, Benson JM. Delayed care with harmfulhealth consequences—reported experiences from national surveysduring Coronavirus Disease 2019.JAMA Health Forum. 2020;1(12):e201463

Birkmeyer JD, Barnato A, Birkmeyer N, Bessler R, Skinner J. TheImpact of the COVID‐19 pandemic on hospital admissions in theUnited States: study examines trends in US hospital admissionsduring the COVID‐19 pandemic.HealthAff. 2020;39(11):2010‐2017

Jeffery MM, D'onofrio G, Paek H, et al. Trends in emergency de-partment visits and hospital admissions in health care systems in 5states in the first months of the CoViD‐19 pandemic in the US.JAMA Intern Med. 2020;180(10):1328‐1333

L46: what was the estimated rate of foregone care and how higher was it in marginalized communities?

L53: You only cite 1 study here; how is the data mixed if 6-7 find the same thing and only one does not? In addition, the cited study that did not find similar association analyzed only Medicare beneficiaries who are different anyway. I would suggest reviewing these more carefully and updating the text

L53-55: This sentence should either be excluded or mentioned way earlier in the document (ie 1st paragraph)

L63: During the pandemic?

L69-71: The authors make an argument that nationally representative survey data do not employ robust sampling strategies and lack representation of such population. Please review this and rephrase. Also, add references so that the reader can judge which studies you are referring to. The study by Giannouchos et al use UI HPS data which are among the most comprehensive nationwide surveys – which the authors have missed in their search. Same throughout L71-73.

L75-79: Similar with the previous comment, I am not sure how under-resourced communities in Baltimore are nationally representative of other similar communities in New Mexico for example. I would suggest the authors rethink their contribution to the literature and the advantage of their study compared to previous work. Maybe the use of a survey with more recent data relative to what is published is a stronger argument.

L83: Adults as 18+ or 18-64?

L84 & 134: What was the rationale for using person-years? I am still not convinced by this. The question / outcome is dichotomous, they weren’t asked how many times – in which case enrollment length would matter. Did the authors consider or do a robustness check on this?

L86: Where these data collected for different studies? If so, your study is retrospective and secondary and you pulled your data from these studies. This should be made more explicit.

TABLE 1 should be an appendix; it is not a finding nor something that the researchers did for this study

L119: So how many were excluded? My understanding is that around 30% were excluded. If so, did the authors check on missingness? Was it at random?

L120: How many in each exclusion?

I see that this addressed in Figure 1 and the first section of the results. Maybe this should be moved in the methods.

L125: Is it possible to identify who skipped, missed, or delayed as separate outcomes? Skip is very different than delay.

L139-140: I don’t understand this. Do you mean stratified analyses?

L145-147: This is very problematic. The study N should be robust and justified, not change throughout.

L149: Why not chi2 or Fisher’s exact?

L139-151: The standard would be to conduct some sort of multivariable regression model to assess variation in foregone rate by various characteristics/factors. I am surprised that the authors did not do this, so I would encourage them to do so. Otherwise, descriptive findings and significances reported can be misleading. This is a serious issue.

L168: To my comment on representation above, 96% are not Hispanic – so your sample has external validity issues. Hence, I would encourage rethinking the strength and contribution of your work in the Introduction

Table2: I would suggest adding 2 columns and comparing those with and without reported foregone care and their p-values here which is how most studies present their findings as in this field

L185-188: I am concerned about this; see my comment above

TABLE 3: The Ns of Total Events are not consistent. This needs to be fixed. This table is confusing as well and suggest updating it as mentioned above

Lack of multivariable regression is a problem here. There is a need for this to explore which associations hold once adjusted for.

L219: An N of 2,000 is not large

L220-227: This might be a strength but is not suitable here. The first paragraph in the discussion should wrap up the findings and provide a comprehensive summary of the main findings. If the authors would like to highlight the strengths of the two samples, this should be mentioned later and more succinctly.

L229: First, how does the identified rate align with the available literature? That is the first big finding.

L238: I am not sure that his result will hold in a regression model

L243 & 251: Reference?

L251: Could this finding be sample specific?

L261: But you didn’t study immunocompromised nor present evidence about those. In addition, among the finding, are any of those at increased risk of mortality? Otherwise, this is a true comment, but not backed up but the study and irrelevant

L275: Isnt that shown in Table 3? If not, what are these numbers?

Limitations: Also, its not nationally representative and lacks external validity

L287: This is an assumption and cant be validated nor justified and thus must be excluded

L229-230 & 241: Were these factors associated with foregone care also before the pandemic? What have pre-pandemic studies found? If these were, then the findings are not novel.

Reviewer #2: This study examines forgone health care during the SARS-CoV-2 pandemic. This is a well-written manuscript that addresses a relevant topic, but I have several concerns, which are listed below.

First, I think the findings are not novel enough. Consequences of the pandemic on foregone care and subgroups more likely to forego care have been well described in the literature. I am not sure that a cross-sectional study can improve our knowledge on this topic.

Second, I am not sure that it is relevant to use "person-years" in a cross-sectional study. If the interest is in the pandemic period, then the study should be limited to the relevant periods. In addition, both studies missed the beginning of the pandemic, when the worst levels of missed care were likely to have occurred.

Third, recommendations on strategies to reduce the spread of the virus do not seem useful more than 3 years after the start of the pandemic and while restrictive measures are still in place.

Finally, the conclusions discuss the results, but given the design and results of the study, this should be removed from the paper.

Please include a description of covariates in the methods section.

Reviewer #3: This paper evaluates whether COVID-19 pandemic was associated with increased forgone care. The authors use two surveys on forgone care conducted among the population of adults residing in Baltimore, MD. They find that the rate of forgone care was highest among individuals with disabilities, functional limitations, financial stress, and multiple comorbidities.

Overall, I think this paper is nicely written and utilizes some novel data. I think a few additions to the analysis will strengthen the implications of this study.

1. The authors mention that they drop individuals with missing data for the key outcomes (forgone care) or inclusion criteria (did they need care during COVID). While I understand the need to remove these observations, it raises the possibility of selection bias. Can the authors say something about individuals with missing outcomes? Were they more likely to be from specific demographic groups? Perhaps this can be mentioned in the Limitations.

2. The authors mention that the study participants were sampled from across Baltimore, with oversampling for underrepresented demographic and socio-economic groups. How did the study population compare to Baltimore overall socio-economic and demographic population composition? Can the authors use Census data for Baltimore to compare their study population to Census Baltimore population? This would greatly help with interpreting the results and external validity implications.

3. The number of individuals reporting not being employed in Table 2 is really high (52.3%). This number is not explained by other characteristics reported in Table 2, such as % with disability (15.5), % with college degree (63.2), % lost housing during the pandemic (7.6), % with financial limitations (30.7), etc. Was the employment so low because it was COVID period, or was it so low for some other reason? How does this number compare to the overall unemployment rate in Baltimore in 2020, 2021, 2022? I think this goes back to my previous comment about comparing study population to the overall Baltimore population using an external data source.

4. Can functional limitations be broken down by % physical, mental, or emotional? Is this group overrepresented by physical or emotional limitations, which have different implications on the need for medical care?

5. The authors use certain diseases to calculate the number of comorbidities in Table 3 (footnote 3). Why were these comorbidities selected? Is it because of the study design? Or were these diseases more common among the study participants? How does the list of comorbidities used in the paper compare to comorbidities used to construct the typical comorbidities indices (Charlson, Elixhauser).

6. The Limitations section mentions absence of insurance status in the study. While it will be impossible to know the insurance status of study participants if it was not recorded in the survey, the authors can say something about the overall insurance rate in Baltimore using Census data. This will give the audience at least some clue into potential distribution of insured/uninsured population in Baltimore.

6. PLOS authors have the option to publish the peer review history of their article (what does this mean?). If published, this will include your full peer review and any attached files.

Reviewer #1: No

Reviewer #2: No

Reviewer #3: No

---

## [Author Response · Author response to Decision Letter 0]

8 Jan 2024

We would like to thank the reviewers for their time and insightful comments. As a result of their suggestions, we have opted to include in this manuscript our multilevel logistic regression analysis, which was done as part of a broader study. Thus, the manuscript has changed significantly, as it now focuses on the findings of our regression analysis. Notably, we believe this to be the most “novel” contribution of this work, as we looked at both individual and community level correlates of forgone care. The purpose of our paper was to explore whether there were any community-level correlates (such as internet access) that accounted for forgone care above and beyond any individual level correlates (e.g., age, comorbidity status). To our knowledge, there exist no published manuscripts on the community-level corelates of forgone care during the COVID-19 pandemic. We believe that the addition of this analysis to our paper greatly increases its value to the existing literature. We would also like to note the addition of one author, Dr. Thomas Inglesby, who helped with refining the study design and methodology for the regression analysis as well as helped to edit this revised manuscript.

Please see our comments to the reviewers below.

Reviewer #1: Thank you for allowing me to review this paper. I have major issues with the paper in general, in particular with the methods and analytical pieces of it. I am also not convinced about the novelty and contribution of this study the way the authors frame it. 

Below are some of my comments:

L41: Studies using data from the Urban Institute capture exactly this question; ie Giannouchos TV, Brooks JM, Andreyeva E, Ukert B. Frequency and factors associated with foregone and delayed medical care due to COVID‐19 among nonelderly US adults from August to December 2020. Journal of evaluation in clinical practice. 2022 Feb;28(1):33-42.

-We have added this citation, along with several others that were inadvertently omitted from this manuscript (see citations 8-19). 

L42: Or choice due to less urgent need – some rate of foregone care might actually be a good thing when patients overutilize the healthcare system. So this statement is partially true and should be revised

-This line has been removed from the revised manuscript.

L38-46: There are more US studies looking into foregone/delayed care thus the authors should review the literature more careful or at least acknowledge that there is more than “limited” evidence (L43)

A short search revealed the following:

• Giannouchos TV, Brooks JM, Andreyeva E, Ukert B. Frequency and factors associated with foregone and delayed medical care due to COVID‐19 among nonelderly US adults from August to December 2020. Journal of evaluation in clinical practice. 2022 Feb;28(1):33-42.

• Findling MG, Blendon RJ, Benson JM. Delayed care with harmfulhealth consequences—reported experiences from national surveysduring Coronavirus Disease 2019.JAMA Health Forum. 2020;1(12):e201463 

• Birkmeyer JD, Barnato A, Birkmeyer N, Bessler R, Skinner J. TheImpact of the COVID‐19 pandemic on hospital admissions in theUnited States: study examines trends in US hospital admissionsduring the COVID‐19 pandemic.HealthAff. 2020;39(11):2010‐2017 

• Jeffery MM, D'onofrio G, Paek H, et al. Trends in emergency de-partment visits and hospital admissions in health care systems in 5states in the first months of the CoViD‐19 pandemic in the US.JAMA Intern Med. 2020;180(10):1328‐1333 

-The Giannouchos and Findling studies are now referenced in our paper. However, the Birkmeyer and Jeffery articles look at hospital trends rather than patient reported forgone care. We agree that there are many articles published on health system trends during the pandemic (e.g., hospital admissions, etc.) but less has been published about patient reported delayed or forgone care. However, we have added the Birkmeyer and Jeffery article as citations for the sentence “Much of this evidence consists of studies describing changes in the volume of services rendered, such trends in hospital admissions and emergency department usage” (lines 42-44) as we do note the preponderance of evidence on this topic in our paper.

L46: what was the estimated rate of foregone care and how higher was it in marginalized communities?

-We have added a few exemplars from existing studies to highlight how some studies have found higher rates/odds of forgone care in marginalized communities, but also that the data have at times been contradictory. Please see lines 48-62.

L53: You only cite 1 study here; how is the data mixed if 6-7 find the same thing and only one does not? In addition, the cited study that did not find similar association analyzed only Medicare beneficiaries who are different anyway. I would suggest reviewing these more carefully and updating the text

-This text has been updated to more clearly reflect the existing literature, including clarifying the population sampled for the exemplar studies (see lines 48-62).

L53-55: This sentence should either be excluded or mentioned way earlier in the document (ie 1st paragraph)

-This sentence has been removed.

L63: During the pandemic?

-No, this was evidence published prior to the pandemic. This has been clarified (see lines 82-83).

L69-71: The authors make an argument that nationally representative survey data do not employ robust sampling strategies and lack representation of such population. Please review this and rephrase. Also, add references so that the reader can judge which studies you are referring to. The study by Giannouchos et al use UI HPS data which are among the most comprehensive nationwide surveys – which the authors have missed in their search. Same throughout L71-73.

-This last paragraph has been completely revised. Please see lines 89-95.

L75-79: Similar with the previous comment, I am not sure how under-resourced communities in Baltimore are nationally representative of other similar communities in New Mexico for example. I would suggest the authors rethink their contribution to the literature and the advantage of their study compared to previous work. Maybe the use of a survey with more recent data relative to what is published is a stronger argument.

-This sentence has been removed.

L83: Adults as 18+ or 18-64?

-Adults 18+. This has been clarified (see line 155-157).

L84 & 134: What was the rationale for using person-years? I am still not convinced by this. The question / outcome is dichotomous, they weren’t asked how many times – in which case enrollment length would matter. Did the authors consider or do a robustness check on this?

-People were asked about experiencing forgone care when they completed the survey. However not everyone completed the survey at the same time so had varying amount of time to have experienced forgone care (surveys were completed from Feb. 2021 to Dec. 2022). Those who have longer exposure to the pandemic would have a greater opportunity to experience forgone care at some point compared to those with short pandemic exposure. However, as suggested by the reviewers we have dropped the reporting of rates and now focus on the association of experience of forgone care and individual and community level factors.

L86: Where these data collected for different studies? If so, your study is retrospective and secondary and you pulled your data from these studies. This should be made more explicit.

-We have more explicitly stated that this is a secondary data analysis using parent study data in the revised draft (see lines 99-109)

TABLE 1 should be an appendix; it is not a finding nor something that the researchers did for this study.

-In the revised draft, we have removed this table and have instead chosen to describe the parent studies in a few short paragraphs (see lines111-148). We believe having this information in the text (vs. an appendix) is critical as it provides context for the parent studies, including sampling strategies.

L119: So how many were excluded? My understanding is that around 30% were excluded. If so, did the authors check on missingness? Was it at random?

-In total, 888 individuals were excluded based on our inclusion criteria (e.g., they did not require care). The others that were excluded (n=87 total) because the participant had discordant answers (n=44), data on the outcome of forgone care were missing (n=42), or because they were missing data on ≥7 covariates (n=1). We compared these 87 individuals who were dropped from the dataset to those who remained in the dataset across all demographic covariates to assess for any systemic differences and there were none.

L120: How many in each exclusion? I see that this addressed in Figure 1 and the first section of the results. Maybe this should be moved in the methods.

-This has been added to the methods section (see lines 152-161).

L125: Is it possible to identify who skipped, missed, or delayed as separate outcomes? Skip is very different than delay.

-Unfortunately, given the way both parent surveys were worded, it cannot be further clarified whether the visit was skipped or delayed. We agree it would have been preferable to have data on both skipped and delayed visits separately.

L139-140: I don’t understand this. Do you mean stratified analyses?

-This sentence has been removed in the revised manuscript.

L145-147: This is very problematic. The study N should be robust and justified, not change throughout.

-Thank you for this comment. In the revised draft, we present the results of our multilevel regression, which used an imputed dataset so that a full, robust dataset was used for all analyses.

L149: Why not chi2 or Fisher’s exact?

-We no longer included these analyses in the revised draft based on reviewer feedback.

L139-151: The standard would be to conduct some sort of multivariable regression model to assess variation in foregone rate by various characteristics/factors. I am surprised that the authors did not do this, so I would encourage them to do so. Otherwise, descriptive findings and significances reported can be misleading. This is a serious issue.

-Please see our comment at the beginning of our reviewer response.

L168: To my comment on representation above, 96% are not Hispanic – so your sample has external validity issues. Hence, I would encourage rethinking the strength and contribution of your work in the Introduction

-We agree that lack of representation of the Hispanic population is a limitation of our study. We have noted this in our limitations section, including that we were unable to incorporate this variable into our final model because of lack of power.

Table2: I would suggest adding 2 columns and comparing those with and without reported foregone care and their p-values here which is how most studies present their findings as in this field

-Table 2 now includes a column for those who reported any type of forgone care and a column for those who did not report forgone care. We did not include p-values in this table, as the significance of each of these variables are tested in the regression analyses using the imputed dataset.

L185-188: I am concerned about this; see my comment above

-We are not quite sure what the reviewer is referencing here…perhaps the description of person-years. We have now changed the focus of our paper to a multilevel regression (vs. person-years using rates) so we hope we have addressed this comment. 

TABLE 3: The Ns of Total Events are not consistent. This needs to be fixed. This table is confusing as well and suggest updating it as mentioned above

-This table has been removed.

Lack of multivariable regression is a problem here. There is a need for this to explore which associations hold once adjusted for.

-This revised draft now focuses on our multilevel regression analysis.

L219: An N of 2,000 is not large

-This sentence has been removed.

L220-227: This might be a strength but is not suitable here. The first paragraph in the discussion should wrap up the findings and provide a comprehensive summary of the main findings. If the authors would like to highlight the strengths of the two samples, this should be mentioned later and more succinctly.

-The discussion section has been revised and the first paragraph now focuses on our primary finding, which was that none of the community-level factors was found to be significantly correlated with forgone care.

L229: First, how does the identified rate align with the available literature? That is the first big finding.

-The discussion section has been revised and we compare some of our findings to the existing literature (see lines 374-378, lines 389-397, lines 411-412).

L238: I am not sure that his result will hold in a regression model

-Our analysis now describes our regression analysis.

L243 & 251: Reference?

-These sentences have been removed.

L251: Could this finding be sample specific?

-These sentences have been removed.

L261: But you didn’t study immunocompromised nor present evidence about those. In addition, among the finding, are any of those at increased risk of mortality? Otherwise, this is a true comment, but not backed up but the study and irrelevant

-These sentences have been removed.

L275: Isnt that shown in Table 3? If not, what are these numbers?

-These sentences have been removed.

Limitations: Also, its not nationally representative and lacks external validity

-This has been added to the limitations (see lines 457-463).

L287: This is an assumption and cant be validated nor justified and thus must be excluded

-These sentences have been removed.

L229-230 & 241: Were these factors associated with foregone care also before the pandemic? What have pre-pandemic studies found? If these were, then the findings are not novel.

-Please see prior comments.

Reviewer #2: This study examines forgone health care during the SARS-CoV-2 pandemic. This is a well-written manuscript that addresses a relevant topic, but I have several concerns, which are listed below.

First, I think the findings are not novel enough. Consequences of the pandemic on foregone care and subgroups more likely to forego care have been well described in the literature. I am not sure that a cross-sectional study can improve our knowledge on this topic.

-Please see our comment at the introduction to our response to reviewers.

Second, I am not sure that it is relevant to use "person-years" in a cross-sectional study. If the interest is in the pandemic period, then the study should be limited to the relevant periods. In addition, both studies missed the beginning of the pandemic, when the worst levels of missed care were likely to have occurred.

-This study now focuses on our regression analysis and not on the rate of forgone care by person-years.

Third, recommendations on strategies to reduce the spread of the virus do not seem useful more than 3 years after the start of the pandemic and while restrictive measures are still in place.

-Thank you for this comment. We have completely re-written the discussion section and have included two paragraphs that discuss strategies for addressing forgone care during future public health crises.

Finally, the conclusions discuss the results, but given the design and results of the study, this should be removed from the paper.

-The conclusions have been re-written.

Please include a description of covariates in the methods section.

-This has been added.

Reviewer #3: This paper evaluates whether COVID-19 pandemic was associated with increased forgone care. The authors use two surveys on forgone care conducted among the population of adults residing in Baltimore, MD. They find that the rate of forgone care was highest among individuals with disabilities, functional limitations, financial stress, and multiple comorbidities.

Overall, I think this paper is nicely written and utilizes some novel data. I think a few additions to the analysis will strengthen the implications of this study.

1. The authors mention that they drop individuals with missing data for the key outcomes (forgone care) or inclusion criteria (did they need care during COVID). While I understand the need to remove these observations, it raises the possibility of selectio

---

## [Decision Letter · Decision Letter 1]

16 Feb 2024

PONE-D-23-27005R1An Evaluation of the Impact of Social and Structural Determinants of Health on Forgone Care during the COVID-19 Pandemic in Baltimore, MarylandPLOS ONE

Dear Dr. Meyer,

Thank you for submitting your manuscript to PLOS ONE. After careful consideration, we feel that it has merit but does not fully meet PLOS ONE’s publication criteria as it currently stands. Therefore, we invite you to submit a revised version of the manuscript that addresses the points raised during the review process.

We look forward to receiving your revised manuscript.

Kind regards,

Tae-Young Pak, Ph.D.

Academic Editor

PLOS ONE

Journal Requirements:

Reviewers' comments:

Reviewer's Responses to Questions

**Comments to the Author**

1. If the authors have adequately addressed your comments raised in a previous round of review and you feel that this manuscript is now acceptable for publication, you may indicate that here to bypass the “Comments to the Author” section, enter your conflict of interest statement in the “Confidential to Editor” section, and submit your "Accept" recommendation.

Reviewer #3: (No Response)

2. Is the manuscript technically sound, and do the data support the conclusions?

Reviewer #3: Partly

3. Has the statistical analysis been performed appropriately and rigorously? 

Reviewer #3: Yes

4. Have the authors made all data underlying the findings in their manuscript fully available?

Reviewer #3: Yes

5. Is the manuscript presented in an intelligible fashion and written in standard English?

Reviewer #3: Yes

6. Review Comments to the Author

Reviewer #3: Thank you for addressing my comments. I would like to push back on authors’ claim that they cannot use Census data to compare the unemployment rate in their population with the overall Baltimore unemployment rate during the same time period. First, the authors can use both the civilian labor force participation rate and unemployment rate. Census also provides numbers for percent of individuals who might not be in the labor force because they are students or retirees. While, it is true that Census numbers include population 16 years and older, the authors do not provide evidence why the numbers for population of 16 years and older should be significantly different from the numbers for population of 18 years and older. Overall, I think a more careful look at Census numbers will allow the authors to compare their (very high) unemployment rate with Baltimore unemployment rate. There is about 20 percentage points difference between the unemployment rate in their sample and the CSA-level unemployment rate. This should be mentioned in the limitations.

7. PLOS authors have the option to publish the peer review history of their article (what does this mean?). If published, this will include your full peer review and any attached files.

Reviewer #3: No

---

## [Author Response · Author response to Decision Letter 1]

11 Mar 2024

We would like to thank Reviewer 3 for taking the time to review our revised manuscript. Please see our response to their comment below. We would also like to note the addition of two authors, whom were inadvertently left off the previous submission. These authors assisted with data acquisition and logistics for the C-Forward parent study. We regret this omission and appreciate the opportunity to add them as authors to the manuscript.

Reviewer #3: Thank you for addressing my comments. I would like to push back on authors’ claim that they cannot use Census data to compare the unemployment rate in their population with the overall Baltimore unemployment rate during the same time period. First, the authors can use both the civilian labor force participation rate and unemployment rate. Census also provides numbers for percent of individuals who might not be in the labor force because they are students or retirees. While, it is true that Census numbers include population 16 years and older, the authors do not provide evidence why the numbers for population of 16 years and older should be significantly different from the numbers for population of 18 years and older. Overall, I think a more careful look at Census numbers will allow the authors to compare their (very high) unemployment rate with Baltimore unemployment rate. There is about 20 percentage points difference between the unemployment rate in their sample and the CSA-level unemployment rate. This should be mentioned in the limitations.

Thank you for this comment. 52.3% of our sample (age 18 years +) reported not being employed full or part-time, meaning they fit into one of these other categories: retired, keeping house, student, laid off, looking for work, or disabled. As you noted, the overall unemployment rate for Baltimore City (age 16 years+) is 29.0%, per the Baltimore Neighborhood Indicators Alliance (note, this data is from 2021). Census data from 2022 indicates that 62.0% of the Baltimore City population was in the civilian labor force (age 16 years+), so we can approximate that 38% were not in the labor force. We suspect that our unemployment rate in our sample was higher due to greater representation of individuals from CSAs with higher unemployment rates. In particular, the CoVPN parent study recruitment methodology included a greater emphasis on persons in lower SES neighborhoods. We have added a sentence addressing this in the limitations section.

---

## [Editor Report · Decision Letter 2]

27 Mar 2024

An Evaluation of the Impact of Social and Structural Determinants of Health on Forgone Care during the COVID-19 Pandemic in Baltimore, Maryland

PONE-D-23-27005R2

Dear Dr. Meyer,

We’re pleased to inform you that your manuscript has been judged scientifically suitable for publication and will be formally accepted for publication once it meets all outstanding technical requirements.

Kind regards,

Tae-Young Pak, Ph.D.

Academic Editor

PLOS ONE
---

## [Editor Report · Acceptance letter]

30 Apr 2024

PONE-D-23-27005R2 

PLOS ONE

Dear Dr. Meyer, 

I'm pleased to inform you that your manuscript has been deemed suitable for publication in PLOS ONE. Congratulations! Your manuscript is now being handed over to our production team.

Kind regards, 

on behalf of

Tae-Young Pak 

Academic Editor

PLOS ONE